# Implant-Supported Fixed Partial Dentures with Posterior Cantilevers: In Vitro Study of Mechanical Behavior

**DOI:** 10.3390/ma16206805

**Published:** 2023-10-22

**Authors:** Fernando García-Sala Bonmatí, Naia Bustamante-Hernández, Jorge Alonso Pérez-Barquero, Jesús Maneiro-Lojo, Carla Fons-Badal, Carla Labaig-Caturla, Lucía Fernández-Estevan, Rubén Agustín-Panadero

**Affiliations:** 1Department of Dental Medicine, Faculty of Medicine and Dentistry, Universitat de València, 46010 Valencia, Spainsusomaneiro@hotmail.com (J.M.-L.); carla.fons@uv.es (C.F.-B.); ruben.agustin@uv.es (R.A.-P.); 2Department of Stomatology, Faculty of Medicine and Dentistry, University of Basque Country, 48940 Leioa, Spain; naiabustamante@gmail.com

**Keywords:** mechanical behavior, cyclic loading, compressive loading, implant-supported fixed partial denture, posterior cantilever

## Abstract

Rehabilitation with dental implants is not always possible due to the lack of bone quality or quantity, in many cases due to bone atrophy or the morbidity of regenerative treatments. We find ourselves in situations of performing dental prostheses with cantilevers in order to rehabilitate our patients, thus simplifying the treatment. The aim of this study was to analyze the mechanical behavior of four types of fixed partial dentures with posterior cantilevers on two dental implants (convergent collar and transmucosal internal connection) through an in vitro study (compressive loading and cyclic loading). This study comprised four groups (*n* = 76): in Group 1, the prosthesis was screwed directly to the implant platform (DS; *n* = 19); in Group 2, the prosthesis was screwed to the telescopic interface on the implant head (INS; *n* = 19); in Group 3, the prosthesis was cemented to the telescopic abutment (INC; *n* = 19); and in Group 4, the prosthesis was cemented to the abutment (DC; *n* = 19). The sets were subjected to a cyclic loading test (80 N load for 240,000 cycles) and compressive loading test (100 KN load at a displacement rate of 0.5 mm/min), applying the load until failure occurred to any of the components at the abutment–prosthesis–implant interface. Subsequently, an optical microscopy analysis was performed to obtain more data on what had occurred in each group. Results: Group 1 (direct screw-retained prosthesis, DS) obtained the highest mean strength value of 663.5 ± 196.0 N. The other three groups were very homogeneous: 428.4 ± 63.1 N for Group 2 (INS), 486.7 ± 67.8 N for Group 3 (INC), and 458.9 ± 38.9 N for Group 4 (DC). The mean strength was significantly dependent on the type of connection (*p* < 0.001), and this difference was similar for all of the test conditions (cyclic and compressive loading) (*p* = 0.689). Implant-borne prostheses with convergent collars and transmucosal internal connections with posterior cantilevers screwed directly to the implant connection are a good solution in cases where implant placement cannot avoid extensions.

## 1. Introduction

Prior to the development of implant dentistry, tooth replacement was possible by means of fixed prostheses supported on the adjacent teeth of the edentulous gap. When the number or quality of the natural teeth was not favorable for the construction of a fixed prosthesis, tooth replacement was performed by fitting partially or completely removable prostheses [1,2]. These prosthodontic treatment alternatives have evolved with the introduction of implantology as an increasingly common treatment that allows for the replacement of missing teeth. The problem is found in situations where we want to rehabilitate our patients and, for various reasons, we cannot position the implants in all the areas we would like, thus having to design prostheses with cantilevers in order to occlude with the antagonist teeth. In such cases where biomechanical stress can generate mechanical and biological problems, the ideal implant or prosthodontic design is unknown [3].

When fabricating an implant-borne reconstruction, several clinical and laboratory aspects must be analyzed, but it will ultimately be the clinician who decides on the type of restoration to be used in each situation [4]. When talking about implant-supported prosthetic rehabilitation, two main groups can be distinguished according to retention: cemented and screw-retained. A cemented prosthesis takes advantage of the power of the cement for retention; crown stability is provided by the cement but is also influenced by the conicity of the abutment because a certain degree of convergence in its walls can increase retention [5]. On the other hand, a screw-retained prosthesis is one that bases its retention on the screw that fixes the reconstruction to the prosthetic abutment [6,7]. Both methods are valid and have advantages and limitations; therefore, it is always necessary to individualize the case and choose the most appropriate retention method for each patient.

According to the definition in the *Glossary of Oral and Maxillofacial Implants*, a complication is an abnormal and unexpected change from the normal treatment outcome [8]. A general distinction is made between biological complications and technical or mechanical complications [9]. Although screw-retained restorations have had good results, many dentists work with cemented prostheses; screw-retained prostheses have a great deal of evidence for their ease of retrievability and reduced biological complications such as peri-implantitis [10,11]. Cemented prostheses, on the other hand, can boast optimal occlusal design, better esthetics, and passivity, while retrievability is related to the use of temporary cements [12].

Implant prostheses are subjected to two types of forces, which are different but related to each other:

Compressive loadings are constant forces, even when there is no occlusal load, which are given by the preload of the prosthetic screws and the absence of passive adjustment. However, when there is no passivity, a type of load is produced that is applied slowly, thus not causing vibratory or dynamic effects on the structure, but is increased gradually from zero to its maximum value, remaining constant.

Cyclic loadings depend on the occlusion force, both functional and parafunctional, so they are inconstant by definition. They are applied when a movement is generated. They can have various forms, impact loads, and fluctuating loads. There are also cyclic loads that are characterized by the repetition of a continuous load.

The aim of this in vitro study was to analyze the mechanical behavior of four types of fixed partial dentures with cantilever posterior extensions on two implants with convergent transmucosal collars by comparing the results of the different groups: in Group 1, the prosthesis was directly screwed to the implant (DS); in Group 2, the prosthesis was indirectly screwed to the implant (INS); in Group 3, the prosthesis was indirectly cemented (INC); and in Group 4, the prosthesis was directly cemented (DC). All the groups were subjected to load tests (compressive and cyclic), and the possible structural alterations (fracture, plastic deformation, loosening, and debonding) of the prosthesis–implant complex were observed by optical microscopy.

The working hypothesis raised in the research was that prostheses with a direct screw and those screwed to the telescopic interface on the implant head would have a worse mechanical behavior after cyclic and compressive loading then cemented restorations, regardless of whether they rested on the abutment or on the convergent machined neck of the implant.

## 2. Materials and Methods

Seventy-six specimens were manufactured on 152 implants (Prama^®^ Sweden and Martina SPA, Padova, Italy) of 11.25 mm × 4.25 mm with a 2 mm deep hexagonal internal connection of 3.4 mm and a convergent transmucosal collar characterized by a cylindrical portion of 0.8 mm and a hyperbolic portion of 2 mm in height.

Each sample consisted of two implants and a CAD-CAM structure with a posterior cantilever that was 1.5 times the length of the anteroposterior distance of the framework design that could be cemented or screw retained to the implants. The study differentiated between 4 groups (Figure 1): Group 1 (DS): 19 CAD-CAM frameworks directly screwed to the implants; Group 2 (INS): 19 CAD-CAM frameworks bonded to titanium interface abutments and then screwed to the implant head telescoping 0.5 mm to the convergent collar of the transmucosal implant; Group 3 (INC): 19 CAD-CAM frameworks cemented on telescopic titanium abutments without a termination line, with the prosthesis resting at 0.5 mm on the machined implant collar; Group 4 (DC): 19 CAD-CAM frameworks cemented on prefabricated titanium abutments without a termination line and the prosthesis resting on the abutment.

To perform the dynamic fatigue tests, the models were designed following the specifications of the UNE-EN ISO 14801:2008 standard [13]. This standard requires the specimens to be mounted in an epoxy resin with a modulus of elasticity of 3 GPa or higher, and peri-implant bone loss must be simulated, leaving 3 mm of the implant surface exposed from the neck down in a coronal-apical direction. To be able to place the implants in a standardized and reproducible position, a positioning key was designed (Figure 2). For this purpose, the test tube was digitized using an extraoral scanner with active triangulation (E3, 3Shape^®^, Copenhagen, Denmark) (Figure 3), and the key was designed with the DentalSystem software (System 2020 version 20.1, 3Shape^®^, Copenhagen, Denmark), which facilitated the positioning of the implants by controlling parallelism and angulation. Once designed, it was resin printed (FREEPRINT Splint 385; Detax, Ettlingen, Germany) using an AsigaMax^®^ 3D printer (Sydney, Australia).

The implants were then screwed to the key and positioned over the test tube, which was filled with Exakto-Form^®^ epoxy resin (Bredent GmbH and Co. KG, Senden, Germany). After curing, the positioning key was removed, and scan bodies (Sweden and Martina^®^) were placed on the implants to digitize the sample. After digitization, the structures of the four groups were designed using Exocad^®^ software (DentalCAD 3.1 Rijeka, Exocad GmbH, Darmstadt, Germany) (Figure 4a,b).

The STL files obtained after the design of the structure were sent to the Echo CAD CAM Milling Center (Padova, Italy), where 76 milled structures in cobalt chromium were manufactured.

A second key per group was then manufactured to position the different prosthetic structures (Figure 5) and thus obtain a good passive fit. The structures of Groups 1 and 2, as well as the titanium abutments of Groups 3 and 4 were screwed to the implants by applying a torque of 25 Ncm with an ISD900 torque-controlled prosthetic screwdriver (NSK, Nakanishi, Japan) by the same operator (FGS) (Figure 6a,b). The cemented structures of Groups 3 and 4 were cemented with RelyX™ Unicem 2 (3M ESPE, Saint Paul, MN, USA).

In this study, the cyclic loading phase was first performed on 12 specimens from each of the groups under study to assess the influence of this type of test on the final result. The cyclic loading tests were performed with the chewing simulator machine (Chewing Simulator CS-4.2 economy line^®^; DS Mechatronik GMBH, Feldkirchen-Westerham, Germany) at the facilities of the Faculty of Medicine and Dentistry of the University of Valencia; a vertical load of 8 kg (80 N) with a vertical movement of 2.5 mm, a horizontal movement of 2 mm, and a speed of 60 mm/s was applied with a steel ball fixed to the mobile axis of the machine (Figure 7). The load was applied to the designed area of the cantilever of the structure for 240,000 cycles, which, according to the ISO standard parameters, corresponds to 1 year of work.

At the end of the fatigue tests, compressive loading was applied to all the specimens using a Shimadzu^®^ universal test machine (Shimadzu Corporation, Kyoto, Japan) at the facilities of the Polytechnic University of Valencia in the Department of Materials and Mechanical Engineering. The load application arm was perpendicular to the descent trajectory of the 100 kN cell connected to a computer, which was made to coincide with the highest point of the cantilever sphere and whose mission was to receive the load and transmit it to the rest of the system. With micrometric adjustments, the flat load applicator was positioned to the smallest visually appreciable distance without touching the specimen.

A load cell displacement speed of 0.5 mm/min was used, and the load was applied until a change in the prosthetic complex occurred (Figure 8), which can be clearly seen in the graphs. The machine was connected to a computer that, using specific TRAPEZIUM-X version 1.00 software (single serial 942356CA, Shimadzu Corporation, https://www.shimadzu.com (accessed on 18 October 2023)), processed and stored the data obtained during the test. The computer software displayed load–deformation graphs, which made it possible to determine for each of the samples the exact load at which the system failed and the type of mechanical behavior to determine the limiting force for altering the implant–prosthetic structure.

After both tests were performed, a microscopic analysis was carried out with a Leica M125^®^ x4 optical microscope (Leica Microsystems GmbH, Wetzlar, Germany) on each of the samples from each group to determine the type of event that had occurred (fracture, plastic deformation, loosening, or debonding) and the location of the mechanical problem.

The statistical analysis employed inferential analysis using the Shapiro–Wilk test with a significance level of 5% (*p* > 0.05). The analysis of variance model, a 2-way ANOVA, was used to evaluate the load (N) until restoration failure as a function of group and test type. To observe significant differences between pairs of groups, the Bonferroni test was used as a post hoc test; to analyze the strength of the materials, the Weibull model was used to predict the probability of failure of the different types of connection. In the microscopic study, chi-squared tests were used by group and test type.

A prior calculation was made of the sample size necessary for an average difference of 50 N in the average force of 2 groups to be detectable as significant using a t-test with Bonferroni correction with 80% power. The 50 N difference was extracted from Karasan’s study for its 2 most similar groups, with SD = 40 N per group [14]. The confidence level was established at 99.17% (*p* = 0.0083) as it was a post hoc comparison of 2 groups in a study that will include a total of 4 groups. The result of the calculation indicated that at least 18 cases per group were needed.

The F test of the ANOVA model obtained a confidence level of 95% considering an effect size of f = 0.4 (large); a power of 82.3% was achieved to detect if the mean maximum load difference was statistically significant between groups.

## 3. Results

Figure 9 shows the results of the maximum compressive load to failure per group. It can be observed that Group 1 (direct screwed, DS) with 663.5 ± 196.0 N was clearly superior to the other three groups, and these were very homogeneous among themselves—428.4 ± 63.1 N for Group 2 (indirect screwed, INS), 486.7 ± 67.8 N for Group 3 (indirect cemented, INC), and 458.9 ± 38.9 N for Group 4 (direct cemented, DC)—independently of the test mode (either the combined test of cycling loading and compressive loading or only compressive loading), until rupture of one of the components of the sample. The mean load depended significantly on the type of connection (*p* < 0.001), with the DS group being clearly superior to the rest.

In the previous graph, a great variability of load values in Group 1 (DS) can be observed. The load is substantially higher than in the other groups, with values that can reach 1002.8 N due to a very large dispersion of the data (Figure 10); the box concentrates 50% of the cases, and the median is shown by the horizontal line that divides it. The upper and lower edges of the box correspond to the first and third quartiles, below which are 25% and 75%, respectively, of the sample. The “whiskers” extend to values in an acceptable range, above which are the outliers (circles) and extremes (asterisks).

In the previous graphs, we observe the great variability of load values detected in Group 1 (DS). That is to say, although it is true that the load was substantially higher than in the rest of the groups, it is also true that the dispersion of the data was very large. In relative terms, it can be estimated that the variability represents 30% of the mean in Group 1 (DS), while in the other groups, it is approximately 5–20%.

When an analysis was performed to evaluate the contribution of each factor (group and test mode), it was observed that the mean load depended significantly on the type of connection (*p* < 0.001). This difference was similar for all of the test conditions (*p* = 0.836) according to the ANOVA model, which concluded a nonsignificant interaction. The addition of a dynamic test phase at the beginning of the study in the SD group had little influence on the results.

With respect to the Weibull failure probability (Figure 11), it was observed once again that the DS group presented average values of stress of 734.46 N, which was much higher than the rest of the groups; the rest of the groups are located to the left of the plane for stress values (below 500 N). The higher the characteristic stress was, the stronger the group, and the higher the Weibull modulus was, the greater the impact of stress on the probability of failure, i.e., as the load increased, the stress rose considerably, and the probability of failure increased.

In addition to this difference in the position of the probability functions, the difference in the slopes of the probability functions is also very evident. The slope of Group 1 (DS) is the most moderate of all, i.e., the effort must be increased considerably to achieve a significant increase in the probability of failure. In Figure 12, it can be seen that it is necessary to increase the effort from 500 N to 800 N to go from a failure probability of 0.2 to 0.8. On the other hand, Groups 2 (INS) and 3 (INC) exhibit similar slopes, but the slope of Group 4 (DC) is even greater. Therefore, these are connections where failure can occur early, especially when a cemented connection is involved.

The relationship between the load and maximum displacement of the structure was analyzed (Figure 12). Group 1 (DS) had a strong load–displacement relationship, with a steep slope indicating that small increases in displacement corresponded to large changes in maximum load. Group 2 (INS) also presented values with a positive correlation between load and displacement. However, it presented some concentrated experimental data points (failure at 400 N/4 mm) that caused the variation in the slope. Cemented specimens, Groups 3 (INC) and 4 (DC), failed at lower loads, similar to specimens belonging to Group 2.

In the image analysis, the screws showed deformation (90.8%) and fracture (6.6%), and only 2.6% were found to be intact; these data were dispersed in a similar way for all groups, which we will detail below. The abutments could not be evaluated for Group 1 (DS) since it is a single direct screw-retained structure. In the other groups, the abutments showed deformation and debonding of the mesial abutment (63.2%), no deformation and debonding of the mesial abutment (5.3%), and fractures (1.8%). The samples presented deformation in both implants in 53.9%, in one implant in 34.2%, and only 11.8% remained intact. The prosthetic structure did not suffer deformation or fracture.

## 4. Discussion

In vitro studies have limitations that are reduced with a standardized protocol [13]. A positioning key and real titanium implants, not replicas, were used for this purpose to better simulate clinical conditions. However, several authors have worked with replicas, which, being made of aluminum, may present results that cannot be extrapolated to the clinical situation in vivo [6,7,8,9,10,11,12,13,14,15,16,17,18,19,20,21,22,23,24,25].

The use of cyclic loading, thanks to a chewing machine in this work, is particularly suitable to reproduce the oral conditions of mechanical stress on implant prosthesis, abutment, and screw interfaces [17,18]. To obtain standardized results, the application of the load was always performed at the same point on the cantilever using an attachment with a flat surface that adequately distributed the load across the entire specimen.

The applied load was 80 N, which is similar to the force used by Rosentritt [18] and represents a conventional occlusal force. This is an important but not fundamental factor, since although some authors left the cantilevers free of occlusion, they still recorded an important incidence of fractures [24].

Group 1 (DS) was the most resistant, as it is a block complex, and the other groups presented interphases and were very homogeneous in their behavior.

Regarding the results obtained, we observed that Group 1 (DS) supported the highest loads at 663.5 N compared to the other three, which were very homogeneous among themselves, at 428.4 N in Group 2 (INS), 486.7 N in Group 3 (INC), and 458.9 N in Group 4 (DC). The study by Karasan [14] on cantilevers over one or two implants with zirconia or titanium interphases and varied union implant abutment with zirconia and titanium obtained values of 226 N and 601 N, respectively.

In Karasan’s work, as in Gehrke’s [14,25], the authors added variations in the material and concluded that titanium abutments behave better than zirconia abutments. The latter were only indicated in the anterior sector with lower occlusal loads.

Cantilevers, such as those analyzed in this work, increase the probability of failure of the prosthesis–abutment complex, as they increase the stresses of the structures with respect to studies performed without cantilevers [26,27]. However, Romanos et al. [28] conducted a systematic review with the aim of identifying whether distal cantilever prostheses could be used safely. An average prosthetic survival rate of 95% over a follow-up period of 7 years was observed, making them a viable treatment option. The common complications reported were screw loosening and/or porcelain fracture. Da Silva stated that cantilevers are not detrimental to peri-implant marginal bone loss or to the survival of the implant–prosthesis complex, although there are mechanical complications, which are greater in longer cantilevers. Therefore, Mehl recommended short extensions [29,30].

Yilmaz’s work is interesting because while he did use cantilevers in his tests, he used much more flexible materials such as HPP (high resistance polymers). Among these, the most widely used was PEEK (polyether ether ketone), for which he obtained very high results (2610 N) compared to our study. That study had several groups in which the height of the connector and length of the cantilever varied. In the cases where tall rather than wide connectors and short cantilever lengths were used, the best results were obtained, a fact that should be taken into account for the design of such structures [15].

The mechanical complications (Table 1) in this study were concentrated at the level of the screws, abutments, and implants; the prosthetic structures remained intact. The use of implants with a non-narrow platform and diameter led to no fractures in any of them.

The objective of dividing the research samples into four groups was to analyze the different clinical options that this type of implant presents for the preparation of an implant-supported fixed partial prosthesis. Due to the convergent morphology of the transmucosal neck, it is an implant with great versatility since it allows it to be partially covered with the prosthesis. When using a telescopic prosthesis concept, this versatility provides it with greater rigidity in its connection with the implant or in patients who, due to their gingival phenotype, have a thin peri-implant mucosa. According to the literature, this mucosa can be clinically thickened due to the improvement of the emerging prosthetic profile using the Biologically Oriented Preparation Technique (BOPT) [31]. In order to compare these two groups of enveloping prostheses on the transmucosal neck (cemented and screw-retained) with their counterparts without covering this part of the implant, we decided to create two other groups of prostheses adapted to the internal hexagonal platform of the implant. In this way, we could assess whether the telescopic effect of the prosthesis affected the resistance of the prosthesis–implant complex.

The limitations of this research work are those inherent to any type of in vitro study on the resistance of materials in implant prostheses, since it is difficult to simulate all the patients’ conditioning factors that influence the prosthesis in order to extrapolate the results to clinical behavior. At the level of sample analysis, it would be advisable, in future research, to compare this type of group with others of similar morphological design on conventional implants, tissue level type (divergent transmucosal neck), and bone level (on convergent abutments with and without prosthetic finish lines). Additionally, the metallic sample should be compared with different monolithic restorative materials, such as zirconia, to analyze whether the cementation of the different structures on the abutments, for cemented and screwed prostheses, could affect the mechanical behavior of the prosthesis–abutment–implant complex. It would be interesting to evaluate the effect of the type and size of the transmucosal neck on biomechanical stress as well as the use of other materials that are able to absorb more load and do not transmit as much stress to the attachments. Evaluating the prosthetic design of the samples in the telescopic morphology groups externally covering the convergent transmucosal neck would not be possible since this is a unique characteristic of this implant.

## 5. Conclusions

After cyclic loading and a compressive loading test, the mechanical load values of implant-supported restorations with posterior cantilevers with different prosthesis–implant interphases showed optimum results for the restoration of the partially edentulous patient:Direct screw-retained restorations showed the best strength values.Complications were mechanical (deformation, debonding, or fracture) at the prosthetic attachment level (screws or abutments) but did not affect the prosthetic structures.

## Figures and Tables

**Figure 1 materials-16-06805-f001:**
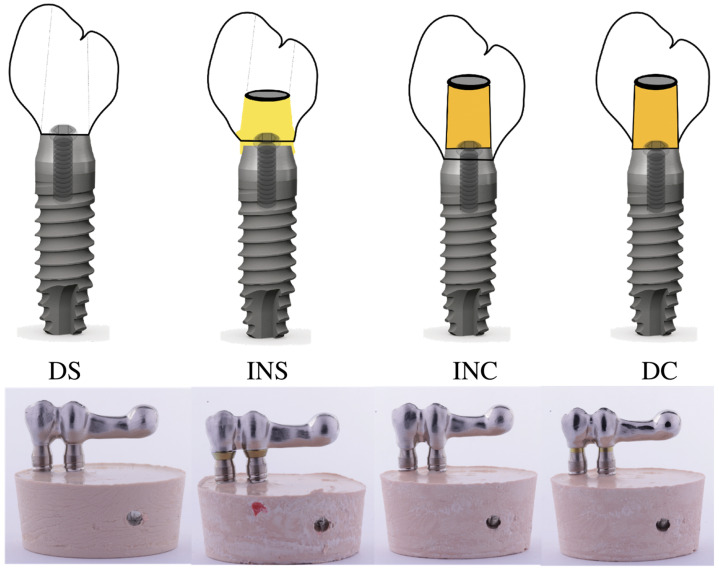
Images of the samples (from left to right): Group 1 (DS), Group 2 (INS), Group 3 (INC), and Group 4 (DC).

**Figure 2 materials-16-06805-f002:**
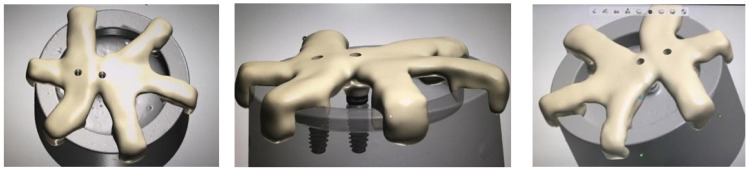
Design of the positioning key for sample standardization.

**Figure 3 materials-16-06805-f003:**
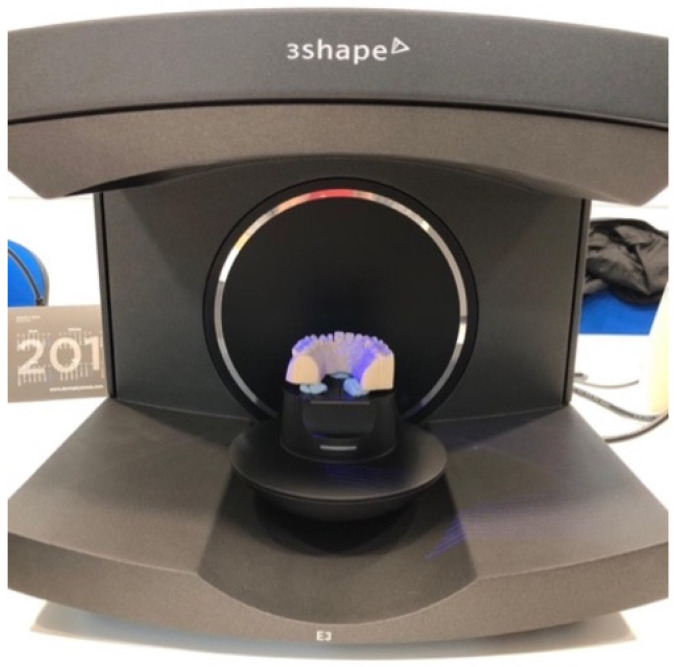
Extraoral scanner used to digitize the samples.

**Figure 4 materials-16-06805-f004:**
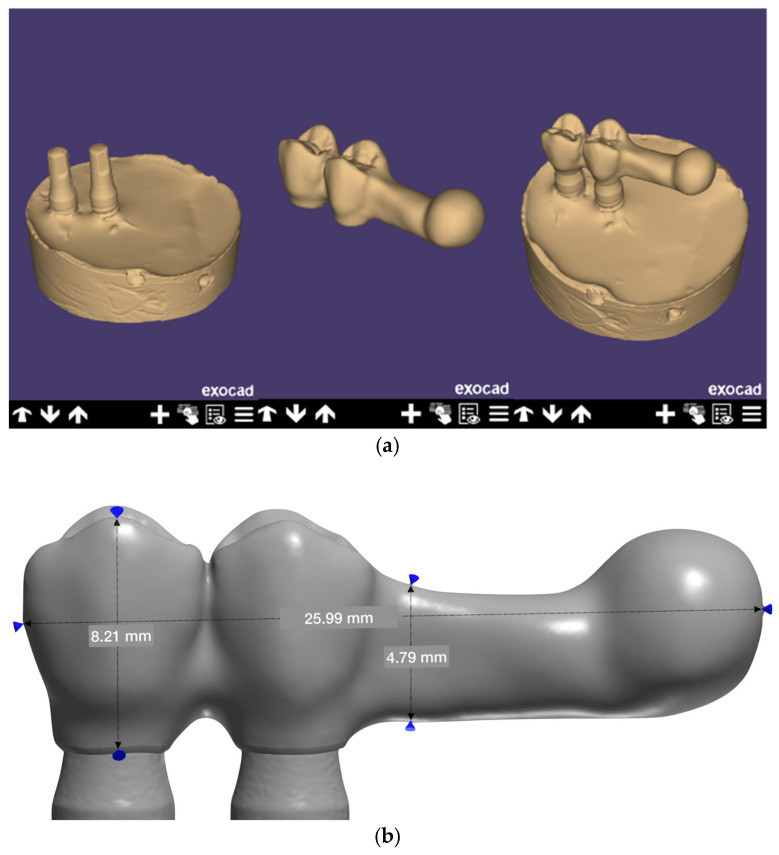
(**a**) Scanning and design of the CAD-CAM structure. (**b**) Dimensions of the implant-supported prosthetic metal structure.

**Figure 5 materials-16-06805-f005:**
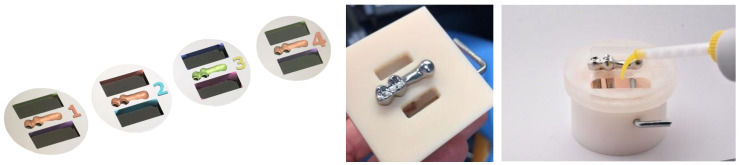
Positioning keys for the assembly of the different structures.

**Figure 6 materials-16-06805-f006:**
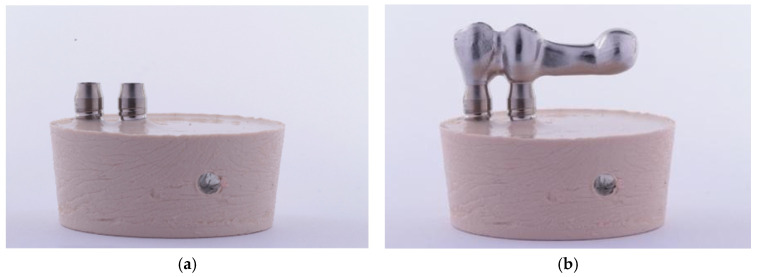
(**a**) Implants positioned after setting. (**b**) Direct screw-retained specimen (Group 1) mounted on the implants.

**Figure 7 materials-16-06805-f007:**
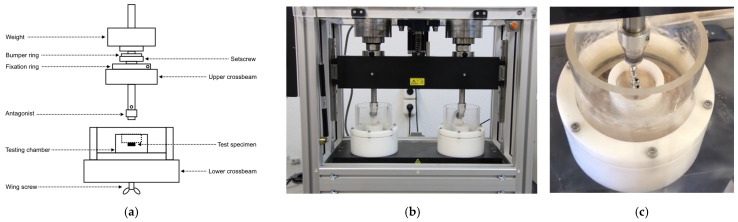
(**a**) Schematic diagram of the chewing simulator machine’s operation and movements. (**b**) Chewing simulator machine. (**c**) Detail of the samples in the chewing simulator machine.

**Figure 8 materials-16-06805-f008:**
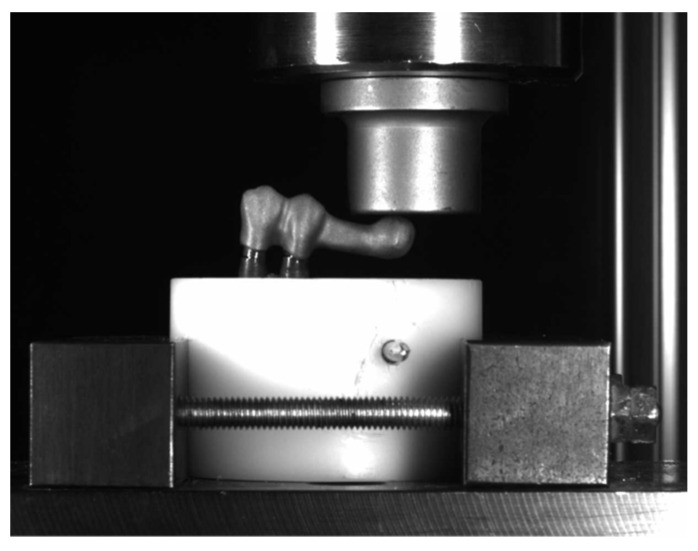
Detail of the sample during the compression test.

**Figure 9 materials-16-06805-f009:**
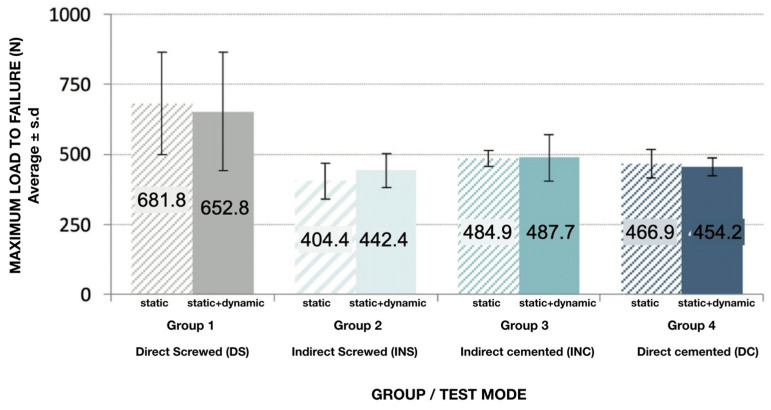
Maximum load to failure (N) expressed as average ±SD according to test group and test mode (**S**: static loading and **S + D**: static and dynamic loading).

**Figure 10 materials-16-06805-f010:**
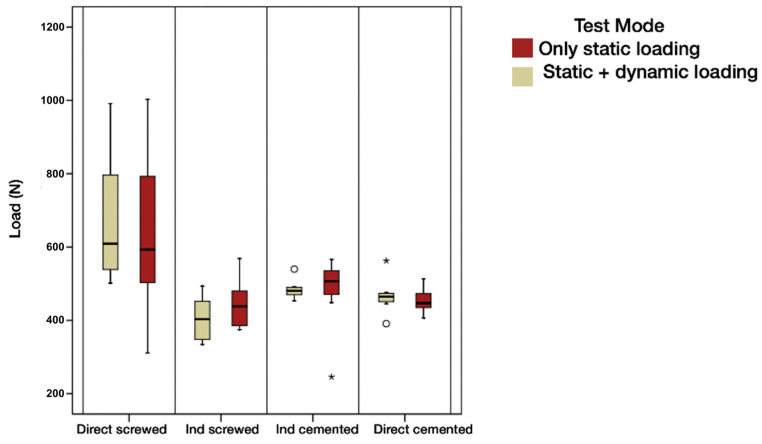
Box plot of maximum load to failure according to test group and test mode.

**Figure 11 materials-16-06805-f011:**
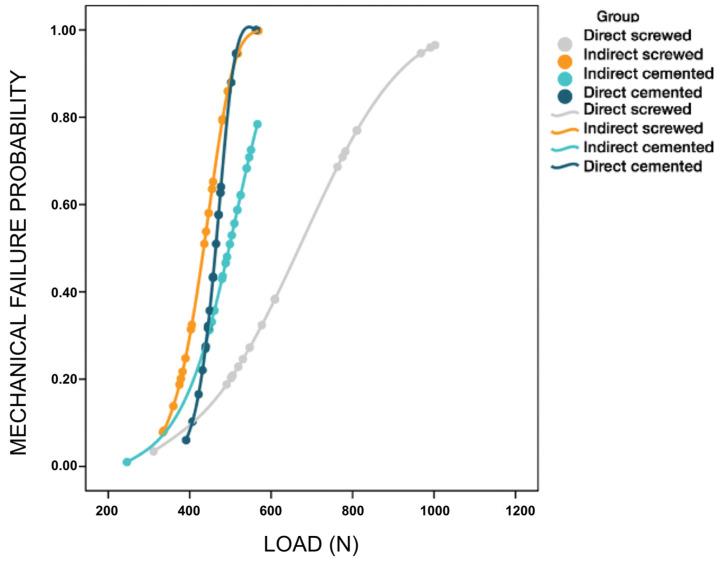
Weibull probability plot.

**Figure 12 materials-16-06805-f012:**
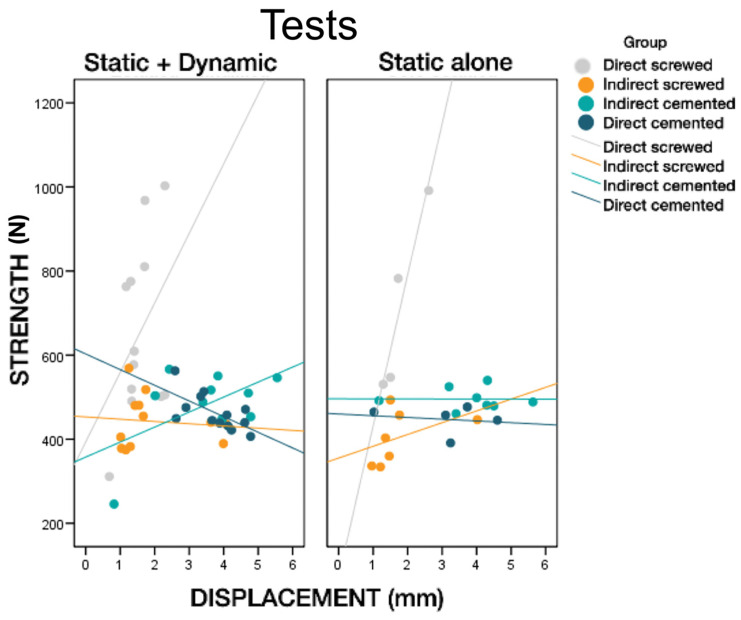
Relationship between displacement and load per group.

**Table 1 materials-16-06805-t001:** Mechanical complications.

COMPLICATIONS	GROUP				
	TOTAL QUANTITY	G1 (DS)	G2 (INS)	G3 (INC)	G4 (DC)
	N	N–%	N–%	N–%	N–%
**SAMPLES**	76	19–100%	19–100%	19–100%	19–100%
**ABUTMENT BROKEN**	1	0	1–5.3%	0	0
**ABUTMENT DEFORMATION**	36	0	3–15.8%	17–89.5%	16–84.2%
**DEBONDING**	18	0	15–78.9%	0	3–15.8%
**SCREW DEFORMATION**	69	17–89.5%	19–100%	17–89.5%	16–84.2%
**SCREW BROKEN**	7	2–10.5%	0	2–10.5%	3–15.8%

## Data Availability

Information is available on request in accordance with any relevant restrictions (e.g., privacy or ethical).

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
