# Peer review of "Implant-Supported Fixed Partial Dentures with Posterior Cantilevers: In Vitro Study of Mechanical Behavior"

_materials, 2023, doi:10.3390/ma16206805_

Round 1
Reviewer 1 Report
Some suggestions/comments are inserted into the attachment.

Minor English improvements are required.
Author Response
Academic Editor.
Materials
Dear Academic Editor:
I’m pleased to resubmit the manuscript of the work entitled, “Analysis of the mechanical behavior of implant-supported fixed partial dentures with posterior cantilever: in vitro study. Part I: Fatigue and compressive loading tests”
Reviewer 1:
Comments and Suggestions for Authors. Some suggestions/comments are inserted into the attachment. peer-review-32461619.v1.pdf
Response: Thank you for your comments, we have edited all the orthographic items and incorporated them into a unique document.
We respond here to some comments
the authors should specify the correlation between the values of the maximum compressive loads mentioned into the text and the values inserted in Figure 10.
Response:
Thank you for your comment. We have edited and clarify the figure 10
The box concentrates 50% of the cases, the median being the horizontal line that divides it. The upper and lower edges of the box correspond to the 1st and 3rd quartiles, below which are 25% and 75% respectively of the sample. The “whiskers” extend to values ​​in an acceptable range, above which are the outliers (circles) and extremes (asterisks).
In the previous graphs we observe the great variability of load values ​​detected in group 1 (AD). That is to say, although it is true that the load is substantially higher than in the rest of the groups, it is also true in the dispersion of the data since it is very large.
the authors are asked to clearly and separately describe somewhere into the manuscript the meaning of the static and dynamic loads, on one hand, and also the static+dynamic loads, on the other hand. additionally, it is important to specify the sequence of the two types of tests: which is the first and which is the second.
Response:
Thank you for your comment. We have explain the test to clarify.
Implant prostheses will be subjected to two types of forces, which are different but related to each other:
Compressive loading are constant forces, even when there is no occlusal load, which is given by the preload of the prosthetic screws and the absence of passive adjustment; However, when there is no passivity, a type of load is produced that is applied slowly, does not cause vibratory or dynamic effects on the structure, but increases gradually from zero to its maximum value, remaining constant.
Cyclic loading: will depend on the occlusion force, both functional and
parafunctional, so they are inconstant, by definition it is that which is applied when a movement is generated. They can have various forms, impact loads, fluctuating loads... There are also cyclic loads that are characterized by the repetition of a continuous load.
this text seems to say that the dynamic test (cyclic loading???) has been performed after the static test which is in disagreement with what the text describes on page 5, rows 135-144.
Response: Thank you for your comment. We have edited the sentence to clarify the order of the test.
the numerical values invoked here should be somehow correlated with those from Figure 12: ca. 1000 N as maximum value for DS group and approx. 570 N as maximum failure load for the other three groups.
Response: Thank you for your comment. We have edited the sentence to clarify the values.
too cloudy; the data should be associated with specific details regarding the samples and the group they belong to.
Response: Thank you for your comment. We have edited the sentence to clarify the values.
Comments on the Quality of English Language Minor English improvements are required.
Response: The certificate of the specialized translator is attached
Reviewer 2:
Comments and Suggestions for Authors
The study is interesting and genuine, however the authors should address the following points to improve the quality of the manuscript:
- The abstract should start with short background statement. The authors should maintain the word limit (please see the authors' guidelines).
Response:
Thank you for your comment. We have edited the abstract to clarify.
Rehabilitation with dental implants is not always possible due to the lack of bone quality or quantity, in many cases due to bone atrophy or the morbidity of regenerative treatments, we find ourselves in situations of performing dental prostheses with cantilevers in order to rehabilitate our patients. thus simplifying the treatment.
- Authors should be expanded to cover the most updated and relevant literature.
Response:
Thank you very much for your appreciation. The discussion has been extended
- Authors should add null hypothesis/hypotheses at the end of the introduction section.
Response:
The direct screw and the screwed on the telescopic interface to the implant head have a worse mechanical behavior, after cyclic and compressive loading then the cemented ones regardless of whether they rest on the abutment or on the convergent machined neck of the implant.
- Please specify the geometry of the internal connection of used implant samples.
Response:
2mm deep hexagonal internal connection
- Why did authors used this implant system specifically since it is not used widely with features that are not similar to the available implants in the market.
Response:
With this type of implant we have the possibility of being able to connect both cemented and screwed abutment at different heights, supported on their connection or also on their transmucosal neck.
- How did the authors determine the sample size?
Response:
A prior calculation was made of the sample size necessary for an average difference of 50N in the average force of 2 groups to be detectable as significant using a t-test with Bonferroni correction with 80% power. The 50N difference was extracted from Karasan's study for its 2 most similar groups, with SD=40N per group. The confidence level was established at 99.17% (p=0.0083) as it was a post-hoc comparison of 2 groups in a study that will include a total of 4 groups. The result of the calculation indicated that at least 18 cases per group are needed
- Why did authors executed two testing mode? (static vs. static-dynamic).
Response:
We did both tesst because we wanted to check if we could find different behaviour after de cycling and not only de static in al the groups
- Was chewing test performed in liquid environment?
Response:
No we didn´t.
- Discussion section is too short. Please expand with comparison between the outcomes with relevant published data. Also add limitations and directions for future research.
Response:
Thank you very much for your suggestion. This has been added to the manuscript at the end of the discussion section
“The limitations of the research work are those inherent to any type of in vitro study of resistance of materials in implant-prosthesis, since it is difficult to simulate all the patient's conditioning factors that influence the prosthesis in order to extrapolate the results to clinical behavior. At the level of sample analysis, it would be advisable, in future research, to compare this type of group with others of similar morphological design on conventional implants, tissue level type (divergent transmucosal neck) and bone level (on convergent abutments with and without termination lines prosthetics) and also compare the metallic sample with different monolithic restorative materials, such as zirconia, to analyze whether the cementation of the different structures on the abutments, for cemented and screwed prostheses, could affect the mechanical behavior of the prosthesis-abutment-implant complex. Evaluating the prosthetic design of the samples in the telescopic morphology groups externally covering the convergent transmucosal neck, their comparison would not be possible since it is a unique characteristic of this implant.”
- Conclusions can be summarized in bullet points.
Response:
Thank you for your comment. We have edited Conclusions.
Reviewer 3:
Comments and Suggestions for Authors
The manuscript entitled 'Analysis of the mechanical behavior of implant-supported fixed partial dentures with posterior cantilever: in vitro study. Part I: Fatigue and compressive loading tests reports the compressive and fatigue testing of implant-supported fixed partial dentures with posterior cantilever. The following points shall be considered for improvement.
- It is recommended to modify the title by shortening the title, highlighting the main highlights of the work.
Response: The title has been edited
Implant-supported fixed partial dentures with posterior cantilever: in vitro study of mechanical behaviour.
- In Figure 1, the above set, it is suggested to provide the dimensions of the specimens for a better understanding.
Response:
- Line 140 - a vertical load of 8 kg (80 N) with a vertical movement of 2.5 mm, a 140 horizontal movement of 2 mm and a speed of 60 mm/s was applied – the criteria for selecting these parameters needs to be incorporated.
Response: The applied load was 80 N, which is similar to the force used by Rosentritt (Rosentritt M., Siavikis G., Behr M., Kolbeck C., Handel G. Approach for valuating the significance of laboratory simulation. J. Dent. 2008;36:1048–1053. doi: 10.1016/j.jdent.2008.09.001.)
and represents a conventional occlusal force. This is an important but not a fundamental factor, since although some authors such Shen left the cantilevers free of occlusion, they still recorded an important incidence of fractures
The machine we use maximum load is 80N and the Shen H., Di P., Luo J., Lin Y. Clinical assessment of implant-supported full-arch immediate prostheses over 6 months of function. Clin. Implant Dent. Relat. Res. 2019;21:473–481. doi: 10.1111/cid.12784
The other parameters used are in accordance with other investigations published before
Selva-Otaolaurruchi EJ, Fernández-Estevan L, Solá-Ruiz MF, García-Sala-Bonmati F, Selva-Ribera I, Agustín-Panadero R. Graphene-Doped Polymethyl Methacrylate (PMMA) as a New Restorative Material in Implant-Prosthetics: In Vitro Analysis of Resistance to Mechanical Fatigue. J Clin Med. 2023 Feb 6;12(4):1269. doi: 10.3390/jcm12041269. PMID: 36835805; PMCID: PMC9960587.
- Line 168 - Microstructural analysis of fracture regions is mentioned, but not present in the results.
Response: we add a table with the microstructural complications
- Figure 8 – right side is not clear. It is recommended to provide clearer images.
Response:
Thank you very much for your appreciation. We have decided to eliminate that part of the low quality image and merge the left part with figure 7.
- Figure 10 – the labelling of Y-axis is missing
Response:
Thank you very much for your appreciation. The figure has been modified to suit your suggestion
- The discussion part is recommended to include more info pertaining the mechanical data including fatigue and compressive strength.
Response:
Thank you very much for your appreciation. The discussion has been extended
Reviewer 4:
Comments and Suggestions for Authors
This is an interesting study where they compared the 4 types of fixed partial 13 dentures with posterior cantilevers on two dental implants (convergent collar and transmucosal 14 internal connection). They found that the direct screw retained to show best mechanical performance. Some comments.
In the Figure 1, it is better to add the full form of abbreviations.
Response:
Thank you very much for your proposal. The figure has been modified according to your suggestion
There is typho in Figure 6.
Response:
The caption has been modified according to your suggestion
Response:
Figures 7 and 8 have been joined according to your recommendation. Thank you so much. This is why we have changed the numbering of figures throughout the manuscript.
It is better to add a Table of all complications in each group.
Response: The table has been added
More discussion is needed the rationale of dividing into 4 groups.
Response:
Thank you very much for your suggestion. This has been added to the manuscript.
“The objective of dividing the research samples into 4 groups was to analyze the different clinical options that this type of implant presents for the preparation of an implant-supported fixed partial prosthesis. Due to this convergent morphology of the transmucosal neck, it is an implant with great versatility since it allows it to be partially covered with the prosthesis, when we wish to use a telescopic prosthesis concept, providing it with greater rigidity in its connection with the implant or in patients who, due to their gingival phenotype, have a thin peri-implant mucosa and we can clinically, according to the literature, thicken it due to the improvement of the emerging prosthetic profile using the Biologically Oriented Preparation Technique (BOPT). (Agustín-Panadero R, Bustamante-Hernández N, Labaig-Rueda C, Fons-Font A, Fernández-Estevan L, Solá-Ruíz MF. Influence of Biologically Oriented Preparation Technique on Peri-Implant Tissues; Prospective Randomized Clinical Trial with Three-Year Follow-Up. Part II: Soft Tissues. J. Clin. Med. 2019, 8(12), 2223). In order to compare these two groups of enveloping prostheses on the transmucosal neck (cemented and screw-retained) with their counterparts without covering this part of the implant, it was decided to create two other groups of prostheses adapted to the internal hexagonal platform of the implant. In this way, it can be assessed whether the telescopic effect of the prosthesis affects the resistance of the prosthesis-implant complex.”
Add limitations of this research.
Response:
Thank you very much for your suggestion. This has been added to the manuscript at the end of the discussion section
“The limitations of the research work are those inherent to any type of in vitro study of resistance of materials in implant-prosthesis, since it is difficult to simulate all the patient's conditioning factors that influence the prosthesis in order to extrapolate the results to clinical behavior. At the level of sample analysis, it would be advisable, in future research, to compare this type of group with others of similar morphological design on conventional implants, tissue level type (divergent transmucosal neck) and bone level (on convergent abutments with and without termination lines prosthetics) and also compare the metallic sample with different monolithic restorative materials, such as zirconia, to analyze whether the cementation of the different structures on the abutments, for cemented and screwed prostheses, could affect the mechanical behavior of the prosthesis-abutment-implant complex. Evaluating the prosthetic design of the samples in the telescopic morphology groups externally covering the convergent transmucosal neck, their comparison would not be possible since it is a unique characteristic of this implant.”
Comments on the Quality of English Language
English improvement should be done to improve the quality of the paper.
Response:
The certificate of the specialized translator is attached The certificate of the specialized translator is attached
Reviewer 2 Report
The study is interesting and genuine, however the authors should address the following points to improve the quality of the manuscript:
- The abstract should start with short background statement. The authors should maintain the word limit (please see the authors' guidelines).
- Authors should be expanded to cover the most updated and relevant literature.
- Authors should add null hypothesis/hypotheses at the end of the introduction section.
- Please specify the geometry of the internal connection of used implant samples.
- Why did authors used this implant system specifically since it is not used widely with features that are not similar to the available implants in the market.
- How did the authors determine the sample size?
- Standard sample preparation protocol was well executed.
- Why did authors executed two testing mode? (static vs. static-dynamic).
- Was chewing test performed in liquid environment?
- Discussion section is too short. Please expand with comparison between the outcomes with relevant published data. Also add limitations and directions for future research.
- Conclusions can be summarized in bullet points.
Author Response

(The authors gave the same response as above.)

Reviewer 3 Report
The manuscript entitled 'Analysis of the mechanical behavior of implant-supported fixed partial dentures with posterior cantilever: in vitro study. Part I: Fatigue and compressive loading tests reports the comprssive and fatigue testing of implant-supported fixed partial dentures with posterior cantilever. The following points shall be considered for improvement.
- It is recommended to modify the title by shortening the title, highlighting the main highlights of the work.
- In Figure 1, the above set, it is suggested to provide the dimensions of the specimens for a better understanding.
- Line 140 - a vertical load of 8 kg (80 N) with a vertical movement of 2.5 mm, a 140 horizontal movement of 2 mm and a speed of 60 mm/s was applied – the criteria for selecting these parameters needs to be incorporated.
- Line 168 - Microstructural analysis of fracture regions is mentioned, but not present in the results.
- Figure 8 – right side is not clear. It is recommended to provide clearer images.
- Figure 10 – the labelling of Y-axis is missing
- The discussion part is recommended to include more info pertaining the mechanical data including fatigue and compressive strength.
Author Response

(The authors gave the same response as above.)

Reviewer 4 Report
This is an interesting study where they compared the 4 types of fixed partial 13 dentures with posterior cantilevers on two dental implants (convergent collar and transmucosal 14 internal connection). They found that the direct screw retained to show best mechanical performance. Some comments.
In the Figure 1, it is better to add the full form of abbreviations.
There is typho in Figure 6.
Figure 7 and 8 can be merged.
It is better to add a Table of all complications in each group.
More discussion is needed the rationale of dividing into 4 groups.
Add limitations of this research.
English improvement should be done to improve the quality of the paper.
Author Response

(The authors gave the same response as above.)

Round 2
Reviewer 1 Report
Only minor suggestion: the authors are asked to consider the measure unit for Load (N) in Figure 11.
Author Response
Reviewer 1
Comments and Suggestions for Authors
Only minor suggestion: the authors are asked to consider the measure unit for Load (N) in Figure 11.
Thank you very much for your suggestion. The manuscript has been modified to incorporate it into Figure 11.
Reviewer 2
Comments and Suggestions for Authors
The authors have addressed the major raised points. The manuscript is recommended for publication after considering the following points.
- Is Fig 13 a table or a figure? If it is a table, change the caption as table.
Thank you very much for your advice. Figure 13 has been modified by table 1.
- In Fig 1, 2, 3, 5, 6, 8- A scale bar should be provided so that readers may get an idea about the size of the specimens.
Thanks for your consideration. The manuscript has been modified by adding a new figure 4b with the dimensions (mm) of the samples studied.
Thank you very much for your comments. For any questions or suggestions, all the authors of the manuscript are at your disposal.
Reviewer 3 Report
The authors have addressed the major raised points. The manuscript is recommended for publication after considering the following points.
- Is Fig 13 a table or a figure? If it is a table, change the caption as table.
- In Fig 1, 2, 3, 5, 6, 8- A scale bar should be provided so that readers may get an idea about the size of the specimens.
Author Response

(The authors gave the same response as above.)
